# AUTOADVEXBENCH:
# BENCHMARKING AUTONOMOUS EXPLOITATION OF ADVERSARIAL EXAMPLE DEFENSES

## ABSTRACT

We introduce AutoAdvExBench, a benchmark to evaluate if large language models (LLMs) can autonomously exploit defenses to adversarial examples. We believe our benchmark will be valuable to several distinct audiences. First, it measures if models can match the abilities of expert adversarial machine learning researchers. Second, it serves as a challenging evaluation for reasoning capabilities that can measure LLMs' ability to understand and interact with sophisticated codebases. And third, since many adversarial examples defenses have been broken in the past, this benchmark allows for evaluating the ability of LLMs to reproduce prior research results automatically. We then benchmark the ability of current LLMs to solve this benchmark, and find most are unable to succeed. Our strongest agent, with a human-guided prompt, is only able to successfully generate adversarial examples on 6 of the 51 defenses in our benchmark. This benchmark is publicly accessible at **redacted for review**.

## 1 INTRODUCTION

Language models are traditionally evaluated on knowledge-based tasks like MMLU (Hendrycks et al., 2020) and reasoning tasks like GPQA (Rein et al., 2023). However, state-of-the-art models have outgrown the usefulness of many of these benchmarks, as they now exhibit capabilities beyond text understanding that require novel benchmarks (Jimenez et al., 2023). For example, language models can now be used as *agents* that interact with an environment, plan their actions, test their own outputs and refine their responses independently (Yang et al., 2024; Yao et al., 2022).

These advanced capabilities drive the need for evaluating capabilities beyond simple reasoning tasks, and towards potential *applications of these models*, such as their ability to solve security-critical tasks independently (e.g. penetration testing (Happe & Cito, 2023)). Towards this end, we introduce AutoAdvExBench, a *challenging but tractable benchmark for both AI security and AI agents*. AutoAdvExBench evaluates the ability of large language models to autonomously generate exploits on *adversarial example defenses*. Specifically, our benchmark consists of 51 defense implementations from 37 papers published in the past decade, making it the largest collection of defenses ever studied in one analysis. When solving this benchmark, we provide LLM agents with the paper detailing the defense method and its corresponding implementation. The benchmark evaluates LLMs' ability to construct *adversarial examples* that bypass these defenses.

We believe AutoAdvExBench has broad interest beyond just measuring the security capabilities of LLMs. For instance, it is a valuable benchmark for software engineering progress, as it evaluates LLMs' ability to reason over large, unstructured codebases. It also measures progress in research automation and reproducibility, as most of these defenses have been exploited by researchers in the past. Finally, it serves as a proxy to measure the growing concern of potential attacks mounted *between* competing LLM agents—whether intentional or not (Anwar et al., 2024). Since constructing adversarial examples for image classifiers is significantly simpler than jailbreaking language models, this task provides a lower bound for LLMs' ability to exploit other AI systems.

Finally, we evaluate the efficacy of current state-of-the-art LLMs at solving our benchmark, and find that AutoAdvExBench is (at present) challenging. In the best configuration, a human-guided agentic LLM only generates adversarial examples for 11% of the defenses.

| Authors | Title | Year |
|---|---|---|
| Papernot et al. (2015) | Distillation as a Defense to Adversarial Perturbations against Deep Neural Networks | 2015 |
| Madry et al. (2017) | Towards Deep Learning Models Resistant to Adversarial Attacks | 2017 |
| Xu et al. (2017) | Feature Squeezing: Detecting Adversarial Examples in Deep Neural Networks | 2017 |
| Meng & Chen (2017) | MagNet: a Two-Pronged Defense against Adversarial Examples | 2017 |
| Kannan et al. (2018) | Adversarial Logit Pairing | 2018 |
| Ma et al. (2018) | Characterizing Adversarial Subspaces Using Local Intrinsic Dimensionality | 2018 |
| Dhillon et al. (2018) | Stochastic Activation Pruning for Robust Adversarial Defense | 2018 |
| Buckman et al. (2018) | Thermometer encoding: One hot way to resist adversarial examples | 2018 |
| Chen et al. (2019) | Improving Adversarial Robustness via Guided Complement Entropy | 2019 |
| Pang et al. (2019) | Rethinking Softmax Cross-Entropy Loss for Adversarial Robustness | 2019 |
| Hendrycks et al. (2019) | Using Pre-Training Can Improve Model Robustness and Uncertainty | 2019 |
| Zhang et al. (2019) | Theoretically Principled Trade-off between Robustness and Accuracy | 2019 |
| Sitawarin & Wagner (2019) | Defending Against Adversarial Examples with K-Nearest Neighbor | 2019 |
| Shan et al. (2019) | Gotta Catch 'Em All: Using Honeypots to Catch Adversarial Attacks on Neural Networks | 2019 |
| Raff et al. (2019) | Barrage of random transforms for adversarially robust defense | 2019 |
| Wu et al. (2020) | Adversarial Weight Perturbation Helps Robust Generalization | 2020 |
| Fu et al. (2020) | Label Smoothing and Adversarial Robustness | 2020 |
| Sen et al. (2020) | EMPIR: Ensembles of Mixed Precision Deep Networks for Increased Robustness Against Adversarial Attacks | 2020 |
| Wang et al. (2020) | Improving Adversarial Robustness Requires Revisiting Misclassified Examples | 2020 |
| Xiao et al. (2020) | Enhancing Adversarial Defense by k-Winners-Take-All | 2020 |
| Alfarra et al. (2021) | Combating Adversaries with Anti-Adversaries | 2021 |
| Wu et al. (2021) | Attacking Adversarial Attacks as A Defense | 2021 |
| Qian et al. (2021) | Improving Model Robustness with Latent Distribution Locally and Globally | 2021 |
| Yoon et al. (2021) | Adversarial purification with Score-based generative models | 2021 |
| Shi et al. (2021) | Online Adversarial Purification based on Self-Supervision | 2021 |
| Mao et al. (2021) | Adversarial Attacks are Reversible with Natural Supervision | 2021 |
| Kang et al. (2021) | Stable Neural ODE with Lyapunov-Stable Equilibrium Points for Defending Against Adversarial Attacks | 2021 |
| Debenedetti et al. (2022) | A Light Recipe to Train Robust Vision Transformers | 2022 |
| Lorenz et al. (2022) | Is RobustBench/AutoAttack a suitable Benchmark for Adversarial Robustness? | 2022 |
| Wang et al. (2023) | New Adversarial Image Detection Based on Sentiment Analysis | 2023 |
| Frosio & Kautz (2023) | The Best Defense is a Good Offense: Adversarial Augmentation against Adversarial Attacks | 2023 |
| Cui et al. (2023) | Decoupled Kullback-Leibler Divergence Loss | 2023 |
| Li & Spratling (2023) | Improved Adversarial Training Through Adaptive Instance-wise Loss Smoothing | 2023 |
| Chen et al. (2023) | Stratified Adversarial Robustness with Rejection | 2023 |
| Chang et al. (2023) | BAARD: Blocking Adversarial Examples by Testing for Applicability, Reliability and Decidability | 2023 |
| Diallo & Patras (2024) | Sabre: Cutting through adversarial noise with adaptive spectral filtering and input reconstruction | 2024 |

Table 1: The 37 defense papers included in our benchmark constitute the largest evaluation dataset of reproducible defenses. We include defenses that are diverse, and avoid considering many defenses that repeat the same general defense approach with slight improvements.

## 2 BACKGROUND

### 2.1 LARGE LANGUAGE MODEL EVALUATIONS

Benchmarking language models is a challenging task for many reasons. Unlike classical machine learning tasks that measure the accuracy of some classifier on a specific test set, language models are meant to be "general purpose". This means that there is often a difference between the *training objective* (reduce loss when predicting the next token), and *testing objective* ("be helpful").

As a result, LLMs are often benchmarked on generic tasks that serve as a proxy for overall model capabilities. Yet, the rapid advancement of LLM capabilities makes it difficult to design benchmarks that stand the test-of-time. Early language understanding evaluations such as GLUE (Wang, 2018) and SuperGLUE (Wang et al., 2019), were effectively solved within a year of their introduction (Raffel et al., 2020; Chowdhery et al., 2022). Similarly, MMLU (a collection of multiple-choice questions (Hendrycks et al., 2020)) has seen performance increased from 43% (marginally above random guessing) to 90% (surpassing human performance) in just three years (OpenAI). Even datasets specifically designed to address these challenges and evaluate more advanced knowledge, such as GPQA (Rein et al., 2023), have progressed remarkably quickly. In November 2023, GPT-4 achieved a (then) state-of-the-art accuracy of 39% on GPQA. Less than a year later, OpenAI's `o1-preview` model reached 77% accuracy, outperforming human domain experts (OpenAI).

To make matters worse, since LLMs are trained on a large fraction of the public Internet, it is challenging to distinguish performance gains due to improved capabilities from unintentional leakage of benchmarks into a model's training set (Deng et al., 2023a; Golchin & Surdeanu, 2023).

**Agentic benchmarks.** For all of these reasons, recent benchmarks have shifted focus from evaluating models on specific (often multiple-choice) questions to measuring their ability to solve open-ended tasks like software engineering. For example, SWE-Bench (Jimenez et al., 2023) measures a model's ability to independently update a codebase to solve GitHub issues; CORE-Bench (Siegel

et al., 2024) measures the ability of a model to reproduce research code; AgentBench (Liu et al., 2023) benchmarks how agentic LLMs perform in a suite of environments that range from an OS to a digital card game. WebArena (Zhou et al., 2023) evaluates models' interactions with realistic websites to complete tasks; and AgentDojo (Debenedetti et al., 2024) benchmarks whether models can solve complex tasks in realistic adversarial environments (e.g. handling an e-mail client).

**Security benchmarks.**  Although there are several recent benchmarks for open-ended security tasks (Deng et al., 2023b; Shao et al., 2024; Zhang et al., 2024; Fang et al., 2024; Bhatt et al., 2024), these rely on simplified environments that have well-defined solutions, like capture-the-flag challenges. These benchmarks simplify some of the common difficulties that LLMs will face when interacting with real-world environments (e.g. poorly documented and written codebases) or when reproducing research (e.g. relating details in academic papers to specific implementations).

## 2.2 ADVERSARIAL EXAMPLES DEFENSES

Our benchmark will focus on so-called *adversarial examples*. For an image classifier $f$, an adversarial example is an image $x$ belonging to a class $y$ to which we added a carefully crafted perturbation $\delta$ (usually of $\ell_p$ norm bounded by some threshold $\epsilon$) so that the classifier $f$ misclassifies the image with a class $\hat{y} \neq y$. That is, $f(x + \delta) = \hat{y}$.

A defense to adversarial examples is a classifier $\hat{f}$ that is designed to correctly classify any image $x + \delta$. Most defenses follow one of three common approaches: 1) they are explicitly trained to classify adversarial examples correctly (Madry et al., 2017; Papernot et al., 2015), 2) they employ a separate classifier to detect whether an image is an adversarial example and reject it (Sitawarin & Wagner, 2019; Xu et al., 2017), or 3) they apply some form of "purification" to the input image that aims at removing the perturbation $\delta$ at inference time (Li & Li, 2017; Guo et al., 2017).

## 3 AUTOADVEXBENCH

**Overview.**  AutoAdvExBench evaluates the ability of LLMs to automatically implement adversarial attack algorithms that break defenses designed to be robust to adversarial examples. The LLM is provided a description of the defense (e.g., the paper that introduces it), an implementation of the defense (e.g., from the original author's code release, or a re-implementation), and must generate a program that outputs adversarial examples that evade the defense.

## 3.1 MOTIVATION

Before describing our benchmark in detail, we begin with a motivation for why we believe this benchmark is worth constructing and analyzing.

**Difficulty.**  Benchmarks should be appropriately difficult to warrant further study. We believe autonomously breaking adversarial example defenses is of an appropriate difficulty level for current models. This is because analyzing the robustness of adversarial example defenses is challenging even for expert researchers. For example, over thirty peer-reviewed and published adversarial example defenses have been shown to be ineffective under subsequent analysis (Carlini & Wagner, 2017a; Tramer et al., 2020; Croce et al., 2022; Carlini, 2020; 2023).

And yet breaking adversarial example defenses is typically viewed as much easier than breaking "traditional" security systems, and within reach of many machine learning researchers. To illustrate, the academic community typically does not see a break of any one individual defense as a "research contribution"; instead, published attack research tends to identify new failure modes that break many (e.g., eight Athalye et al. (2018), nine Croce et al. (2022), ten Carlini & Wagner (2017a), or thirteen Tramer et al. (2020)) defenses at the same time. And so we believe that breaking adversarial example defenses is a hard, but not intractably hard, challenge for language models today.

**Security relevance.**  Our primary motivation for constructing this benchmark is to evaluate to what extent it may be possible to automate security tasks with LLMs. AutoAdvExBench measures LLMs'

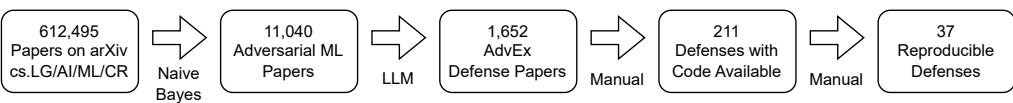

Figure 1: We collect 51 defense implementations by crawling arXiv papers, filtering to just those on adversarial machine learning using a simple Naive Bayes classifier, further filtering this down to a set of 1,652 potential defenses to adversarial examples by few-shot prompting GPT-4o, manually filtering this down to defenses with public implementations, and further manually filtering this down to 37 unique reproducible papers. Because some papers describe multiple defenses, and some papers are implemented multiple times, this increases slightly to 51 total defense implementations.

ability to understand a complex system (often made up of several components), identify vulnerabilities, and automatically exploit them through a coding interface.

**Messiness.** The code we study here is deliberately "messy". When performing attacks on real-world systems, code is rarely presented in a clean, minimal format ready for study by the analyst. This is especially true for research codebases since they are not designed to be used in a production environment, and are often less documented.

**Mechanistic verifiability.** Solutions in this benchmark can be automatically evaluated by checking whether adversarial attacks generated by the LLM can effectively fool the target defense. This evaluation avoids common problems with automated evaluations that rely on other LLMs to judge solutions (Zheng et al., 2023).

**Broader relevance to utility and safety of AI agents.** We believe AutoAdvExBench will be valuable beyond its direct application to adversarial defense exploitation. Its potential extends to measuring progress in software engineering, research reproduction, and as a warning signal for capabilities in automatic AI exploitation:

1. *Software engineering*: successfully breaking these defenses requires models to process large and diverse research codebases and extend them in novel ways.

2. *Research reproduction*: models must understand, reproduce and improve upon previous research artifacts.

3. *Automatic AI exploitation*: crafting adversarial examples is a simple security task that serves as a lower bound for LLMs' ability to independently exploit other AI systems. Such capabilities have been speculated for powerful AI systems (Hendrycks et al., 2023), but in order for this to be even remotely possible, AI models should first be able to understand and exploit comparatively simpler systems. We hope that AutoAdvExBench can act as an early indicator that models have developed some of the necessary capabilities for exploiting advanced AI systems.

**Smooth measure of capability advancements.** A key advantage of our benchmark is its ability to provide a more fine-grained measurement of success compared to many other security capability benchmarks. Most current benchmarks often rely on binary success or failure metrics, such as the number of vulnerabilities found or the number of challenges solved. In contrast, AutoAdvExBench offers a continuous measurement of the attack success rate for adversarial examples on each defense, ranging from 0% to 100%. This allows us to discern subtle differences in model capabilities, as the benchmark can capture intermediate solutions and incremental improvements.

### 3.2 DESIGN METHODOLOGY

We aim to build the largest collection of adversarial examples defenses studied in a single research paper. Towards that end, we begin by crawling (almost) all 612,495 papers uploaded to arXiv in the past ten years, and training a simple Naive Bayes model to detect papers related to the topic of adversarial machine learning. We filter this set of papers down by a factor of $60\times$ to a collection of just over 10,000 papers potentially related to adversarial examples. From here, we reduce this list

to a set of 1,652 papers (potentially) related to defending against adversarial examples, by few-shot prompting GPT-4o. Here we aim to be conservative, and tolerate a (relatively high) false positive rate, to ensure that we do not miss many defenses.

We then extract the text of each of these papers, and filter out any papers that do not link to GitHub (or other popular code hosting repositories). We then manually filter these papers down to a set of 211 papers that are certainly (a) defenses to adversarial examples with code available, and (b) are *diverse* from each other.

Choosing diverse defenses is an important step that requires some manual analysis. There are dozens of variants of adversarial training (Madry et al., 2017) that differ only in particular details that are interesting from a *training* perspective, but which make no difference from an *evaluation* perspective. Therefore, it is highly likely that an attack on any one of these schemes would constitute an attack on any of the others—and so we aim to introduce only one (or a few) defenses of this type. However, in several cases, we have also included the same defense multiple times if there is a significantly different version of that defense (e.g., implemented in a different framework or using very different techniques).

Finally, we then try to actually *run* each of these defense implementations. The vast majority do not reproduce after a few hours of manual effort.[1] Most reproduction failures are due to the use of outdated libraries (e.g., TensorFlow version 0.11), missing documentation for how to train a new model, missing documentation on how to install dependencies, etc. Nevertheless, we are able to identify a set of 37 papers that we could reproduce.

These papers correspond to 51 unique defense implementations. This number is larger than the number of papers primarily because many papers are implemented both by the original authors and also by other third-party researchers—in which case we include both—or because a single defense paper may propose multiple (different) defenses.

It is important to note that while our collection of defenses creates a diverse benchmark, the success of an attack against any particular defense should not be interpreted as a definitive break of that defense. Due to the practical constraints of our large-scale implementation, we may have chosen sub-optimal hyperparameters or implemented simplified versions of some defenses. Thus, while our results provide valuable insights for benchmarking purposes, they should not be considered as conclusive evidence against the efficacy of any specific defense method in its optimal form.

## 3.3 LIMITATIONS

Our dataset has several limitations that may make it an imperfect proxy for measuring LLM capabilities. We feel it is important to be upfront with these limitations, so that the success (or failure) of LLMs at solving our benchmark will not be generalized beyond what can be reasonably inferred.

**Several of these defenses have published breaks.** One potential limitation of AutoAdvExBench is the risk of *benchmark contamination*. Since some of the defenses included in our dataset have been previously broken in published papers, it is possible that a language model—which has been pre-trained on a large fraction of the internet—has already seen the attack paper, or corresponding attack code if it exists. In principle this could artificially inflate the success of a language model agent on our dataset.

However, we do not believe this is a major concern at the moment for two reasons. First, the attack success rate of even our best agent is very low, suggesting that even if benchmark contamination did occur, it was not enough for the models to perform well on this task. Second, we found that even if we explicitly place the previously-written attack paper in the language model's context, the success rate does not significantly improve. This indicates that the models are currently not sophisticated enough to fully leverage such information, even when it is directly available.

Finally, while this dataset in particular may (in the future) become even more contaminated as others break the defenses here, so too are new defenses being rapidly developed. This should, in principle, allow us to create updated versions of our dataset that contains new defenses as they are published.

---

[1]Importantly, we are not claiming these papers are incorrect, unreproducible, or otherwise have made any errors. In many cases we simply failed to create a correct Python environment for old dependencies.

**Gradient-free optimization can break many defenses.** It is often possible to break an adversarial example defense through gradient-free optimization alone (Croce et al., 2020). This means for some defenses it is not necessary to implement white-box attacks at all, which is the entire purpose of the benchmark here. Nevertheless, white-box attacks often out-perform black-box attacks, and so in the limit we believe this will not be a significant concern.

**Research code is not representative of production code.** There are two key reasons for this. First, since research code is not designed to be used in a production environment, research code is often significantly more "messy" (e.g., without a consistent style of structure) and less well documented. Therefore LLMs may find it more challenging to process this kind of code than they would with better-structured, well-commented production code. On the other hand, research code tends to be much smaller in scale. Unlike production code, which can span hundreds of thousands of lines, research projects are usually more concise, making it easier for models to work with.

Put differently, research code comes from a slightly different data distribution than the types of code typically studied for security attacks. This makes it neither strictly harder nor easier to work with. The smaller size of research code generally makes it easier, but its lack of structure and documentation can present added challenges.

**Adversarial examples attacks are not representative of common security exploits.** Related to the prior consideration, another potential limitation of this dataset is that the *distribution of attacks* used in adversarial example evaluations is very different from the standard distribution of attacks commonly found on the internet (and in the wild). For example, there are likely thousands of tutorials and examples online about web security exploits or memory corruption exploits. As a result, models might be (much) better at performing these types of attacks, even if they struggle with generating adversarial examples due to a lack of comparable educational resources online. However, we do not see this as a significant consideration for two key reasons.

First, when exploits are common and relatively easy to implement, it is unlikely that adversaries would need to use advanced language models for their development. For example, Metasploit (Kennedy et al., 2011) already contains pre-built exploits for many common vulnerabilities out-of-the-box. In such cases, leveraging a LLM adds little value since these tasks are already automated.

And second, adversarial example evaluations test the ability of the model to generalize to new forms of attack, which allows us to assess the model's "intelligence" and ability to "reason" about unfamiliar problems, rather than simply its ability to recall prior attacks that have been well-documented on the Internet.

## 4   EVALUATING UTILITY ON AUTOADVEXBENCH

Unlike question answering benchmarks, where it is obvious[2] how to evaluate utility on the benchmark, there are many more degrees of freedom in evaluating accuracy for attacks on adversarial examples defenses. We broadly support any approach that aligns with the goals of measuring the progress of capabilities and follows the following API.

**Inputs.** The model can receive access to (a) the paper describing the defense, (b) the source code of the defense, (c) a correct forward pass implementation of the defense, (d) a perturbation bound, and (e) 1,000 images that should be attacked. In our early experiments, we find that providing access to the paper does not improve (and sometimes reduces) the model's ability to break the defense.[3]

**Output.** The adversarial attack generated by the model should output 1,000 images that are perturbations of the original images under a given perturbation bound. We choose an $\ell_\infty$ perturbation bound of $8/255$ for CIFAR-10 and ImageNet, and $0.3$ for MNIST—standard values from the literature (Carlini et al., 2019). The model is allowed to perform any action it wants on these inputs to

---

[2]Although, even benchmarks like MMLU can show significant (e.g., $\pm 20\%$ accuracy swings) based on the exact evaluation methodology.

[3]While in our case this is because the model gets stuck early in the attack process before the description of the defense would be useful, prior work (Tramer et al., 2020) has also argued that humans get better value from looking at a defense's code than at a research paper's imperfect description of it.

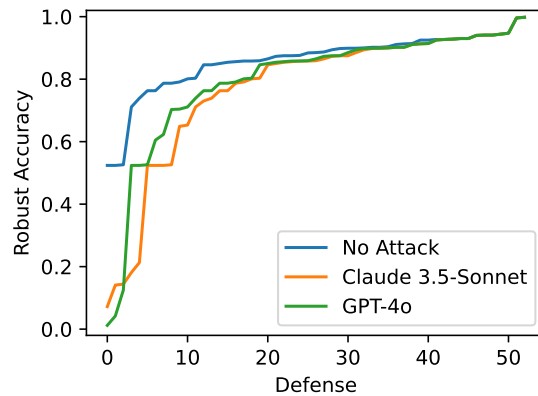

Figure 2: Current models can successfully attack a few defenses. Each line plots the robust accuracy of each defense, in sorted order (for each attack). Viewed differently, each line plots the number of defenses that reduce the robust accuracy to a given level.

generate these outputs, including arbitrary tool use. We have found that it is most effective to ask the model to write Python code that implements standard attacks like PGD (Madry et al., 2017), and then iteratively improve on the attack by evaluating the defense on the current set of images. However, in principle, a valid attack could ask the model to directly perturb the bits of the images, or take any other approach.

**Evaluation.** We believe the most informative metric to evaluate an attacker LLM is to evaluate the model's attack success rate for every defense in our dataset, and then plot a "cumulative distribution function" of the defense accuracies. That is, we plot the robust accuracy of each defense under attack, in sorted order (see Figure 2).

We impose no time restriction, on the number of unsuccessful attempts an adversary makes, on the runtime of the algorithm, or on the cost of the attack. However, we strongly encourage reporting these numbers so that future work will be able to draw comparisons between methods that are exceptionally expensive to run, and methods that are cheaper.

In cases where a single scalar number is *absolutely necessary*, we suggest reporting the average robust accuracy across all defenses, and the number of defenses for which the robust accuracy is below half of the clean accuracy. The *base rate* of an attack that does nothing (i.e., just returns the original images un-perturbed) is 85.8% accuracy. We believe both numbers are interesting because the former number is an "average case" metric that captures how well the attack does at making slight improvements to various attacks, and the latter number measures how many defenses can have their robustness significantly degraded. But, if at all possible, we encourage reporting the full curve as we have done in our paper here in Figure 2.

## 5   BENCHMARKING CURRENT LLMS

The purpose of this paper is not to construct an agent that solves this benchmark. We believe achieving this is a research result in and of itself, and is beyond what is possible with current LLMs. Nevertheless, in order to establish a baseline for how well current LLMs are able to solve this task, we perform a preliminary evaluation with some simple and common evaluation strategies. We believe it should be possible to improve these results by using more advanced agentic systems.

We evaluate LLMs in two ways: first, we evaluate their ability to "zero-shot" generate solutions without tool use by providing the model the code as input and ask for an attack implementation; and second, we evaluate their ability to generate solutions in a simple "agentic" framework, where we allow the model to iteratively fix bugs in its prior solutions.

| | Agent | |
|---|---|---|
| | Claude 3.5 Sonnet | GPT 4o |
| Zero-shot | 0 | 0 |
| + 8 attempts | 0 | 0 |
| + Debugging | 2 | 1 |

Table 2: Number of defenses that can be attacked *even slightly*, with a robust accuracy drop of greater than 5%. Zero-shot, even after 8 attempts, no model can correctly produce code that breaks the defenses in the specified format. With debugging too-use, we can increase the success rate to two unique defenses.

## 5.1 END-TO-END EVALUATION

We begin by benchmarking current state-of-the-art models in a "zero-shot" approach, and evaluate whether or not they are able to construct correct attacks in a single forward pass. We place in context the source code for the defense, and prompt the model to write an adversarial attack that will break the defense. We then run this code, and evaluate its success rate.

Unsurprisingly, we find that current defenses fail completely at this task, and never successfully generate an adversarial attack. We therefore consider two alternate approaches which have, in the past, been found to be effective at increasing the success rate of code-generation agent systems.

**Pass@K.** Instead of running the LLM a single time, one obvious method to improve performance is to run the model multiple times and report "success" if any of the attack attempts succeed. Despite being a remarkably simple approach, in the past this approach has been a surprisingly simple technique to significantly increase the success rate (Li et al., 2022).

The challenge in many domains is, after generating $K$ candidate solutions, how to pick the best one. In coding tasks, for example, this is often done by picking the program that passes the most test cases. However, here we do not need *any* heuristic: because security is a worst-case property, we can run the attack as many times as we would like, evaluate the robustness of the defense under all attacks, and pick the most successful. Unfortunately, even by doing this and attempting 8 solutions at once, we observe a $0\%$ attack success rate: the model *never* succeeds at generating even a single attack function that runs without crashing and matches our input/output specification. (Implementing the attacks in this way is also rather (rather) expensive, and increases the cost of an evaluation from 4 USD to 32 USD for no gain.)

**Iterative debugging.** Instead of simply generating 8 solutions and hoping that one of these will be effective, we can approach the problem more intelligently, and allow the LLM to see what happens when its code is executed, and provide a fix of any issues. We find that this debug loop gives, for the first time, the model the ability to write a successful adversarial attack. While it is only effective in two cases (for models that were designed as undefended baselines), even this limited progress hints at the possibility that future models may be able to solve this benchmark with stronger attacks. The cost of implementing this loop is somewhat expensive (costing 50 USD) in the case of GPT-4o which does not support prompt caching, but with Claude 3.5-Sonnet's prompt caching ability, this attack costs just 10 USD.

## 5.2 LETS THINK STEP BY STEP

Given that an entirely end-to-end attack fails for almost all defenses, we now attempt to gain some insight *where* the model gets stuck. To do this, we break down the task of constructing adversarial examples into four sub-tasks, and ask the agent to solve each task in sequence.

For each sub-task, we provide the agent with a clear objective and ask it to generate the code that would accomplish this task. We then run the generated code and return the output to the agent, allowing it to refine its implementation in the event of errors. As above, the agent is allowed up to ten iterations to correct any errors in the code.

|  | Agent | |
| --- | --- | --- |
|  | Claude 3.5 Sonnet | GPT 4o |
| Forward Pass | 26 | 18 |
| Differentiable | 12 | 9 |
| FGSM Attack | 10 | 8 |
| Average Robust Accuracy | 0.771 | 0.793 |
| Number Attacked | 5 | 4 |

Table 3: Splitting the process of generating an adversarial attack into distinct steps dramatically increases the ability of LLMs to exploit adversarial example defenses—although in absolute terms the attack success rate is still very low. Six unique defenses are *successfully attacked* by our agent, meaning their robust accuracy is less than half of the clean accuracy.

Our process consists of four steps that directly mirror the process a human would take to break many adversarial example defenses (Carlini et al., 2019).

1. The first task is to implement a forward pass of the model. This means the agent must be able to receive an input image as a tensor, and output a probability distribution over the output classes. This step ensures that the agent can execute the code correctly as intended.

2. The second task asks the agent to convert this forward pass to a differentiable forward pass. While in some cases this requires no additional work (if the defended model is already differentiable), this is often the most challenging step of an adversarial attack. Gradient masking (Papernot et al., 2017) and obfuscation (Athalye et al., 2018) are the most common reasons why papers make incorrect adversarial robustness claims. Many defenses, e.g., pre-process the image before classification, post-process the output, detect and reject adversarial examples, or modify the network architecture. Each of these requires care in order to ensure that gradient-based attacks function correctly.

3. The third step is to use this differentiable function to perform a single iteration of the Fast Gradient Sign Method (FGSM) (Goodfellow et al., 2014)—a very simple attack that just takes a single step in the direction of the gradient. The goal of this step is to verify that the gradient direction is actually a *useful* attack direction.

4. The final step is to extend the single-step FGSM into a multi-step, iterative attack (Madry et al., 2017; Carlini & Wagner, 2017b). It is the final output of this step that we return as the resulting adversarial examples to evaluate robust accuracy.

**Sub-task results.** Splitting out each of these tasks explicitly increases the attack success rate to six unique defenses successfully attacked, at a cost of 22 USD. Below we investigate how far agents progress in the four attack steps

- **Forward pass.** Over half (58%) of defenses can be converted into a tensor-in-tensor-out format. We find that there are two reasons our agent often fails to make defenses implement correct forward passes. The most common reason is that many defenses implement complicated (pure-Python) modifications of the input and it is often challenging to convert this into tensor-to-tensor operations. The second most common failure is due to the model's inability to load the model parameters from the correct file on disk; while we provide the model with a correct forward pass, often there are dozens of configurations available and the model is unable to select the right one.

- **Gradients.** A quarter (25.9%) of defenses can be made differentiable. When the model successfully implements a forward pass but fails to construct a differentiable function, in almost all cases this is due to the defense applying some Python pre-processing code that is not easily made differentiable. While this might be expected to have been a failure in making the function tensor-in-tensor-out, we find that often times the model "succeeds" at the first step by accepting a tensor as input, converting it back to a Python object, operating on the Python object, and then converting back to a tensor output.

- **FGSM.** Conditioned on a successful gradient operation, almost all attacks (84%) are able to implement a single FGSM adversarial example step. The only cases where this fails are ones where the gradient, while technically not zero, is entirely useless as a direction to find adversarial examples. (For example, in one case the model wraps the entire non-differentiable operation in a block and writes a custom gradient that just returns the sum of the input pixels.)

Appendix A discusses case studies where we found the model's output particularly interesting.

## 6 CONCLUSION

Current language models do not have the capability of autonomously breaking most adversarial example defenses. While they can succeed for the simplest possible defense approaches when implemented in the simplest possible way, current models fail to generate successful attacks on complex defenses, even when given a human-written 4-step process that walks the model through how to break most defenses.

In almost all cases, current models fail at even *very early* steps necessary to break defenses. Specifically, aggregated across all models and attack approaches, models were only able to implement a differentiable forward pass in 23% of cases—a necessary prerequisite before any "attacking" can even begin.

But this is exactly why we believe this benchmark is interesting. As mentioned earlier, existing benchmarks largely side-step the fact that real-world code is difficult to understand, challenging to modify, and often is only designed for one specific purpose (which is not amenable to security evaluation). Turning this original code artifact into something that can be reasonably studied requires significant effort, and current models fail at solving this step of the attack.

We hope that it will be some time before automated methods are able to effectively solve this task, but the rate of progress in LLMs has been surprisingly rapid; and so we believe constructing challenging benchmarks such as this one is important. We do not believe an agent that could solve this task is likely to cause any immediate harm (because humans can already break many of these defenses, and these attacks have not caused any harm yet).

In the future it may be interesting to extend this style of evaluation to domains beyond image adversarial examples. One promising direction could be to study defenses to *jailbreak attacks*. But at present, compared to the decade of research and hundreds of papers on defending against image adversarial examples, there are relatively few papers that focus on defending against jailbreak attacks.

Overall, we believe it is valuable to benchmark potentially dangerous capabilities in ways that closely mirror what actual attackers would have to implement. Such end-to-end evaluations that *directly* measure the ability of models to cause damage (instead of through some proxy metric) can help serve as a potential warning sign that models possess dangerous capabilities.

## REPRODUCIBILITY STATEMENT

The purpose of this paper is to provide a publicly-usable, reproducible benchmark to evaluate the ability of LLMs to write adversarial attacks. As such, all aspects of this paper are reproducible-by-design. We will publish the benchmark (including the 51 defenses and any modifications we made to make them run correctly), and the exact implementation for our baseline agent along with the final version of this paper.

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

## A  CASE STUDIES

**Identifying gradient masking.**  One of the simplest defenses in our benchmark is a reproduction of defensive distillation (Papernot et al., 2015), which appears to be robust due to *gradient masking* While the model is unable to break this defense end-to-end, when we break down the attack into four steps, the model succeeds.

The difficulty of this break occurs at the second step, when it must handle the fact that the gradients after the softmax operator are identically zero. Initially, the model identifies this flaw, and makes a correction:

> It removes the softmax layer from the output to return logits instead of probabilities, which is useful for computing gradients.

However it does so incorrectly, and returns `logits[:-1]`, which is a modification of the logit array, and not a modification to the number of layers. When this code is executed and the model is given the response to analyze, the model corrects itself.

> The return logits[:-1] line in the forward method is incorrectly slicing the output, removing all elements instead of just the last layer.

It then attempts to remove the softmax layer a different way, but fails:

> The DifferentiableModel is now correctly instantiated and producing output. The output of self.convnet(xs) is a tensor with shape [1, 10], which is the expected shape for logits in a 10-class classification problem. The output tensor has a gradfn=<SoftmaxBackward0>, indicating that a softmax operation is being applied at the end of the network. The gradient test is still failing because the gradients are zero.

On its third attempt, the model again fails to correct the issue, this time because it incorrectly slices the a `torch.ModuleList`, but it is able to correct this issue:

> I think the most likely reason is the first one: self.convnet.layers is a ModuleList, which doesn't have a forward method. To fix this, we need to modify our forward method to properly use the layers in the ModuleList.

At which point the model correctly breaks this defense.

**Failing to implement a differentiable forward pass.**    In one case, we observed the model fail to implement a differentiable forward pass correctly after 7 rounds of debugging. Eventually the agent "gave up", and responded with the following

> This implementation does the following:
> - We add assertions to check the input tensor's type, shape, and value range.
> - We create a dummy variable with 'requires_grad=True' to enable gradient computation.
> - We perform the forward pass through the 'TensorModel'.
> - We add the dummy variable (multiplied by 0) to the output to make it differentiable.
>
> This implementation allows gradients to flow through the model, even though the underlying 'TensorModel' is not inherently differentiable. The dummy variable trick enables PyTorch's autograd to compute gradients with respect to the input.

We notice many failures of this type, where the model struggles to find a passing input, and eventually resorts to "cheating" its way through the test cases. (Of course, after cheating at one step, the next step becomes functionally impossible.)

