# OpenReview forum: "AutoAdvExBench: Benchmarking Autonomous Exploitation of Adversarial Example Defenses"
_ICLR.cc/2025/Conference — Submitted to ICLR 2025_

### Official Review · Reviewer_gDHc · 2024-10-27

**Soundness:** 3
**Presentation:** 2
**Contribution:** 2
**Rating:** 5
**Confidence:** 3

**Summary:**

The paper presented AutoAdvExBench, which evaluated if LLM can automatically exploit defenses to adversarial examples. Contributions include:
- evaluates if models can match the abilities of expert adversarial machine learning researchers
- measure LLMs’ ability to understand and interact with sophisticated codebases
- evaluates the ability of LLMs to reproduce prior research results automatically

The results show: the strongest agent, with a human-guided prompt, is only able to successfully generate adversarial examples on 6 of the 51 defenses in the benchmark.

**Strengths:**

- It's an interesting benchmark to evaluate LLMs' capacity to autonomously exploit defenses to adversarial examples.
- The defense implementations are collected comprehensively and rigorously.
- The limitations are fairly presented.

**Weaknesses:**

The presentation can be improved:
- It takes me a long time to find out that adversarial examples are for image classifiers. LLMs also have so-called adversarial attacks- it would be better to make it clear of the specific task in the abstract as well as early in the introduction to make it clear.
- The introduction didn't actually specify what it is "autonomously generate exploits on adversarial example defenses", before jumping into what the benchmark includes and what is its impact.
- The abstract focuses on the potential impact of the benchmark a lot, Yet it shall better inform what is the benchmark is really about-- the first sentence didn't convey it clearly.

As a benchmark paper, more LLMs should be considered and benchmarked. However, we only see the results for Claude 3.5 and GPT-4o.

If that is because other LLMs are so weak for the task, I personally think a good benchmark for current sota LLMs should be able to distinguish their capacity. If most of them fail for the benchmark, then it may not be a good choice at this time.

**Questions:**

See weakness.

---

> ### Author Response · Authors · 2024-11-22
> **Response to Review**
>
> We thank the reviewer for their time and valuable feedback. We address the weaknesses and questions as follow:
>
> > more LLMs should be considered and benchmarked
>
> We have tested the frontier models that rank highest in all coding benchmarks as of the submission date in the paper. Our camera ready will also include results with models that were released recently: o1 (mini and preview) and newer Claude. We attempted other models on this benchmark, but all had a 0% success rate.
>
> Are there other models that the reviewer thinks would be valuable to include? If so we can try to run these as well.
>
> > If all models fail the benchmark, then it may not be a good choice at this time.
>
> We disagree with this statement. A benchmark should have plenty of room for improvement to avoid saturation early after release, as we discuss in the motivation section. Releasing a benchmark that can already be nearly perfectly solved by one or more models is of little value to the community. As a point of reference, the first year ImageNet was released the best model had 50% accuracy, and when MMLU was released, the best model had just 20% accuracy above random chance. Our baseline agent success rate is not considerably far away from these numbers, and demonstrates the utility of this as a benchmark for hopefully several years to come.
>
> > Presentation can be improved
>
> We are improving some of these issues and will update together with the new model results.

---

> > ### Comment · Reviewer_gDHc · 2024-11-24
> > **Thanks for your response.**
> >
> > Thanks for the response. I would maintain my original score as borderline.
> > While the idea of this benchmark is good for me, the information we can get from the current version of the paper is limited.

---

### Official Review · Reviewer_enyd · 2024-10-28

**Soundness:** 2
**Presentation:** 2
**Contribution:** 3
**Rating:** 5
**Confidence:** 4

**Summary:**

The paper introduces AutoAdvExBench, a benchmark designed to evaluate the ability of large language models (LLMs) to attack adversarial example defenses. The benchmark includes 51 defenses from 37 papers, and the LLMs are provided with the research paper and the corresponding code for the forward pass of the defense.
The authors test state-of-the-art LLMs like Claude 3.5-Sonnet and GPT-4o in various scenarios, finding that current models struggle significantly. Zero-shot attempts were entirely unsuccessful, and even with iterative debugging, only a few defenses were successfully attacked.
The paper highlights the challenges LLMs face in automating security tasks, emphasizing their limitations in understanding and exploiting such defenses.

**Strengths:**

1. The authors invested significant effort in filtering relevant papers and collecting reproducible code, resulting in a comprehensive and valuable dataset of 51 adversarial defenses from 37 papers. This rigorous collection process enhances the benchmark's credibility and relevance to the research community.

2. The benchmark setup involves working with "messy" research codebases, which closely reflects real-world challenges in adversarial defense scenarios. This realistic approach adds significant value, as it pushes current LLMs to handle complex, imperfect code environments typical in real applications.

3. The paper employs definitive evaluation methods that do not rely on black-box metrics, such as LLM-as-a-judge approaches. Instead, it uses measurable, transparent performance metrics, ensuring that the results are both interpretable and replicable.

**Weaknesses:**

1. The paper lacks detailed examples of interactions with the LLMs, such as specific prompts used or a complete end-to-end example of at least one of the defenses and its corresponding developed attack stages. Including visual aids or detailed walk-throughs of successful or failed attacks could greatly enhance readability and engagement. The absence of such details makes the content dense leading to a less engaging reading experience.

2. The current experimental settings are overly general for the LLMs, so it's unsurprising to observe such low success rates. From my experience, even for simpler coding tasks, I find that even the best LLMs require significantly more detail—such as an initial code draft, context explanations, and multiple rounds of conversation and reflection, as in a chat scenario—to achieve the desired results. For example, would a 6-turn interaction (interactive human-AI chat session) with these models (like GitHub Copilot settings) yield much better results?

3. The paper could have explored more powerful methods, such as ICL few-shot examples, fine-tuning, or retrieval-augmented generation (RAG), to better evaluate LLMs' capabilities in solving the benchmark. These approaches could have provided valuable insights into the strengths and limitations of LLMs when leveraging different training and interaction methodologies. The current experimental settings are necessary but insufficient.

4. This is a coding task, yet the paper's background lacks information on state-of-the-art LLMs specifically trained for coding. You've tested with GPT-4o and Claude 3.5 Sonnet, but are there other LLMs worth experimenting with?

5. In Section 5.1 (End-to-End Evaluation), in the sentence "Unsurprisingly, we find that current defenses fail completely at this task ..." do you mean "current models"?

**Questions:**

Please refer to the weaknesses.

**Details Of Ethics Concerns:**

The authors have effectively adhered to the ethical concerns.

---

> ### Author Response · Authors · 2024-11-22
> **Response to Review**
>
> We thank the reviewer for their time and valuable feedback. We address the weaknesses and questions as follow:
>
> > Details on the implementation.
>
> We are working to include visual demonstrations of executions in the appendix (they take a lot of space!) as well as creating a project webpage (that we can not link to currently during anonymous review), and we will open-source the implementation of all defenses and our baseline agent for reproducibility.
>
> > The current experimental settings are overly general for the LLMs.
>
> We implement a baseline agent with SOTA models and agree there is room for improvement with more specialized frameworks. However, designing better agents is its own research contribution and is outside the scope of the paper introduces the new benchmark. (Notice, for example, that the ImageNet and MMLU papers did not introduce highly competitive models---they just introduce the benchmark itself.) We highlight some of the limitations we spotted to help with the design of more powerful agents.
>
> > Experiments with specialized code models
>
> We have tested the frontier models that rank highest in all coding benchmarks. Our camera ready will also include results with models that were released recently: o1 and newer Claude. Are there other models that the reviewer thinks we should test?
>
> > Writing clarifications
>
> We will include additional background on LLMs for writing code as suggested. Yes, we do mean *models* in the sentence "we find that current defenses fail completely at this task" and will correct this.

---

> > ### Comment · Reviewer_enyd · 2024-11-25
> > **Thank you**
> >
> > Thanks for the explanation and including some additional details in the paper.
> > I decided to increase my score from 3 to 5.

---

### Official Review · Reviewer_Xffd · 2024-10-30

**Soundness:** 3
**Presentation:** 4
**Contribution:** 3
**Rating:** 6
**Confidence:** 4

**Summary:**

This paper introduces AutoAdvExBench, a benchmark to evaluate if large language models (LLMs) can autonomously break defenses to adversarial examples, given access to defense (a) the paper describing the defense, (b) the source code of the defense, (c) a correct forward pass implementation of the defense, (d) a perturbation bound, and (e) 1,000 images that should be attacked. The authors attempt both an end-to-end approach and a four-step, human-guided approach. Based on the low success rates, the authors conclude that current language models cannot autonomously break most adversarial example defenses.

**Strengths:**

-  An interesting attempt to test LLMs on a well-established security problem, i.e., defenses against adversarial examples. The authors have provided strong motivations for proposing such a new benchmark.

- The paper is written well, clearly stating the contributions beyond the literature and the significance.

- The fact that current LLMs largely fail in solving such a well-defined problem is somewhat surprising and so calls for future work.

**Weaknesses:**

Presentation:

1. The reviewer appreciates that the authors have put much effort into describing the limitations, even as early as in Section 3.3. However, several points in Section 3.3 Limitations and Section 3.1 Motivation are actually from the same perspective but do not well connect. Specifically, as the reviewer understands, although the benchmark is “difficult” and “security-relevant”, ”Adversarial examples attacks are not representative of common security exploits”. Although the benchmark is “Messy”, its “Research code is not representative of production code”. It may help if the authors could first present the perspectives, and then talk about the motivations/advantages and limitations of the benchmark in each perspective.

2. Throughout the paper, only Figure 2 was referred to in the main text, making the more summarized information in tables/figures not helpful for understanding the paper. The authors should explicitly reference relevant tables and figures when discussing results in the main text. This would help readers connect the discussion to the supporting data more easily.

3. The term “successfully attacked” is not defined before use. For example, it appears in the second to last sentence of the Abstract and the caption of Table 3. In addition, it is not clear what the numbers in Table 3 mean.


Experiment:

1. Although it is OK to show current LLMs are not good at solving the new benchmark, there is no exploration of potential ways to improve them. For example, It is interesting that current LLMs still struggle with simple operations, e.g., “models were only able to implement a differentiable forward pass in 23% of cases”. However, the authors have not attempted to solve this and have not even discussed potential solutions. They could include ideas for targeted training and improved prompting strategies.

2. The authors state that “We impose no time restriction, on the number of unsuccessful attempts an adversary makes, on the runtime of the algorithm, or on the cost of the attack. However, we strongly encourage reporting these numbers so that future work will be able to draw comparisons between methods that are exceptionally expensive to run, and methods that are cheaper.” The reviewer does not get why they have not reported these numbers in this paper (which seems not to require too much additional effort).

**Questions:**

Please see the above weaknesses.

---

> ### Author Response · Authors · 2024-11-22
> **Response to Review**
>
> We thank the reviewer for their time and valuable feedback. We address the weaknesses and questions as follow:
>
> > no exploration of potential ways to improve agents
>
> We agree; but as we mention above, benchmark papers are not typically expected to also introduce solutions. The purpose of a benchmark paper is to show that there exists a new task that researchers could work to solve. We expect that it should be possible to design special-purpose agents to improve the attack success rate, but that is a contribution in its own right.
>
> > Reporting additional measures
>
> We will report a complete set of numbers in the paper. On average successful attacks completed in 16 +/- 7 minutes. Unsuccessful attacks took longer and had a higher variance, with a runtime of 22 +/- 10 minutes. We only run each model once (or, when indicated, five times). We report the cost of the attack in the text of the paper. In the camera ready, we can include a table with these results in the appendix.
>
> > Presentation issues
>
> Thank you for identifying these. We are working on those and will update the paper together with clarity improvements, as well as with results from newer o1 and Claude models, and will include these in a final publication version of this paper.
>
> We define successfully attack as "meaning their robust accuracy is less than half of the clean accuracy" (L444). We can move this to be earlier in the paper. The numbers in Table 3 indicate the total number of defenses that our agent can successfully pass for each of these steps. For example, 3.5 Sonnet can implement FGSM for 10 of the defenses.

---

> ### Comment · Reviewer_Xffd · 2024-11-22
> **thanks for the response**
>
> Thanks for the response. I would maintain my original positive score (since increasing it to 8 would be too much considering that the addressed concerns are just about presentations).

---

### Official Review · Reviewer_qJjZ · 2024-11-04

**Soundness:** 3
**Presentation:** 3
**Contribution:** 3
**Rating:** 8
**Confidence:** 4

**Summary:**

This paper proposes an agentic benchmark for LLMs where the task is to automatically break various adversarial robustness defenses. It includes implementations and PDFs for a large number of adversarial robustness defenses, some of which have published vulnerabilities. The goal of the LLM agent is to output 1000 perturbed images under a standard \ell_{\infty} bound that break the defenses.

**Strengths:**

- This is a clever idea for an agentic benchmark. The task is complex but easy to evaluate, and it provides a way to measure how useful LLM agents could be for stress-testing defenses proposed by the ML community, which I think is an interesting future use case of AI agents.
- The work is timely. Multiple new agentic benchmarks have been proposed recently, including SWE-bench and MLE-bench. This paper continues that line of benchmarking work, but with an emphasis on automated stress-testing for adversarial training defenses.
- The writing is clear.
- The benchmark seems well-designed. The task is easy to understand. Useful data was curated to enable the agents to perform the task (paper code and PDFs).
- The baseline evaluations show that the task is tractable.

**Weaknesses:**

For defenses with published vulnerabilities, it would be good to include a check for whether the model is aware of these vulnerabilities or discovers them from scratch. I realize this isn't relevant to current models, since they aren't very good yet, but it could be an interesting thing to check in future models. I see your response in lines 261-262. This is just a comment that the paper would be stronger with proactive measures to address this.

**Questions:**

No questions for now.

---

> ### Author Response · Authors · 2024-11-22
> **Response to Review**
>
> We thank the reviewer for their time and valuable feedback. We address the concern about checking if models were aware of the exploit beforehand as follows.
>
> Most of the defenses covered in the benchmark do not have publicly available exploits. Further, for the ones where papers describing the break are public, we found that providing the paper text in-context does not significantly improve the attack success rate. We believe this is because even for those that do, it is not trivial for agents to identify when and how to use the knowledge to exploit the actual implementation of the defense.

---

> > ### Comment · Reviewer_qJjZ · 2024-11-27
> > **Response**
> >
> > Thanks. My one concern has been addressed. I will keep my score.

---

### Official Review · Reviewer_uSMe · 2024-11-04

**Soundness:** 2
**Presentation:** 2
**Contribution:** 3
**Rating:** 5
**Confidence:** 5

**Summary:**

This paper proposes a benchmark, AutoAdvExBench, to measure large language models' (LLMs) ability to exploit other AI systems. To be specific, AutoAdvExBench evaluates LLMs' ability to construct adversarial examples that bypass the corresponding defense methods. Experimental results demonstrate that AutoAdvExBench is challenging with a low attack success rate: only 6 of the 51 defenses are successfully attacked by their strongest agent.

**Strengths:**

+ Measuring AI's ability to exploit other AI systems is necessary and meaningful for preparing for future safety risks. Automatic AI exploitation becomes feasible due to the automated nature of the AI agent system and could result in catastrophic risks. This paper provides a realistic implementation for such a speculative threat model. Therefore, it will be a good proxy to monitor the progress of AI and prepare for the possible safety risks.

+ The construction process of AutoAdvExBench is solid: this paper collects reproducible and diverse defense papers with manual checks to compose their benchmark, which will provide a strong basis for measuring AI exploitation ability.

**Weaknesses:**

- **The evaluation metrics lack comprehensiveness.** The paper only reports robust accuracy and the number of successful attacks as metrics, neglecting more detailed analyses of the various capabilities agents need to overcome the benchmark. Areas like code comprehension, code completion, long-context understanding, and the novelty of proposed ideas are overlooked. For example, while the paper finds that only a quarter of defenses can be made differentiable (line 480)—a necessary step before designing new attacks—it’s unclear which capabilities agents lack that lead to these limitations. Simply reporting post-attack robust accuracy does not reveal which specific capabilities are bottlenecks, nor does it offer insights into where current models fall short.

- **Data contamination is a significant concern** that the authors have not adequately addressed. They do not provide empirical results to demonstrate the extent of data contamination in their benchmark. Although they state (line 262) that allowing agents access to paper data does not significantly improve success rates, this does not definitively rule out contamination. Furthermore, as the benchmark does not use the most advanced agent frameworks, it’s conceivable that more sophisticated agents could leverage memorized information to generate attack codes if contamination were present, raising questions about the benchmark’s validity and challenge level.

- **The captions of each table and figure lack essential details, making them difficult to follow.** For instance, in Figure 2, it’s unclear how the caption’s statement, "each line plots the number of defenses that reduce the robust accuracy to a given level," corresponds to the curves in the figure. Similarly, in Table 3, the meanings of terms such as "forward pass," "differentiable," and "FGSM attack" and the respective numbers are not fully explained. This also seems inconsistent with the analysis of FGSM in the main text (line 486), where an accuracy rate of 84% is mentioned.

- **The benchmark does not leverage the latest agent frameworks, such as multi-agent systems with specialized roles, which could potentially address the challenges posed by the benchmark.** The coding capabilities of the agents used in the paper seem quite limited (see Table 3), suggesting that the benchmark may not be challenging enough for cutting-edge agent models.

**Questions:**

- **The analysis of diversity among selected defense papers is insufficient.** Diversity is a crucial metric for this benchmark, yet the distinctions between each defense paper are not clearly highlighted (lines 224–230). It would be helpful if the authors could highlight the unique aspects of each defense paper and delineate their differences.

I would like to raise my score if the authors could address my concerns and questions.

---

> ### Author Response · Authors · 2024-11-22
> **Response to Review**
>
> We thank the reviewer for their time and valuable feedback. We address the weaknesses and questions as follow:
>
> > evaluation metrics lack comprehensiveness.
>
> We agree that intermediate steps and isolated abilities could be evaluated. However, one property of a scalable and objective benchmark is using deterministic evaluation metrics that will remain consistent. Robustly evaluating intermediate progress is challenging because there are many ways to game the benchmark. The fundamental property of "robust accuracy" is that it can be computed deterministically and identically by everyone, thus providing an overview of the overall abilities of the models to complete this task autonomously.
>
> However, we do agree with the reviewer other signals could be interesting, especially early on when the overall attack success rate is low. This is the reason why we report partial progress on several tasks. Other papers may choose to report other interesting numbers if they find them interesting. But we ultimately have to quantify the benchmark down to a single number, and we pick "robust accuracy" as that number.
>
> > Data contamination is a significant concern.
>
> As we mention above, dataset contamination is a problem for any benchmark. We have taken multiple steps to measure the degree to which this is the case, and believe that it is not a significant problem at present.
>
> > The benchmark does not leverage the latest agent frameworks
>
> Current agent frameworks are designed to complete specific tasks, not a general "solve anything". As such, there is no framework that can be used for our benchmark directly to the best of our knowledge. We have tried our best to design a baseline implementation using the latest models and following the standard agent process. Although we agree specialized frameworks may be able to improve upon this baseline, these are outside the scope of our work. We are executing the benchmark on the new O1 and Claude models for camera ready and will include these results.
>
> > The analysis of diversity among selected defense papers is insufficient.
>
> We have implemented all defenses with working code in the literature. (And several with non-working code that we corrected.) Each defense is inherently a different task, because in order for the paper to be published, it must have shown novelty from all other prior work. We will remark this in the updated version of our paper.

---

> > ### Comment · Reviewer_uSMe · 2024-12-02
> > **Thanks for your response**
> >
> > Thank you for your reply. However, I believe my concerns have not been adequately addressed, and I have decided to maintain my score. Specifically:
> >
> > 1. **Evaluation comprehensiveness**: I agree that using robust accuracy is a good, consistent metric, but it's also important to carefully examine the bottlenecks that are revealed when the LLM agents deal with the benchmark. For example, what proportion of the defenses can a LLM describe, in natural language, the ideas for breaking these defenses? The answer to this question would provide more information about the agent's ability to design attack methods than robust accuracy.
> >
> > 2. **Data contamination**: My concerns about this point are similar to the ones mentioned above. I would like to know if and to what extent the agents may "remember" how to break the defenses.
> >
> > 3. The authors did not address my confusion regarding the captions of some tables and figures.
> >
> > 4. **Diversity among selected defenses**: I agree that each defense should show novelty compared to prior work, but I believe it would also be meaningful to provide a high-level categorization of the selected defenses. For example, which methods are based on pruning, which are based on adversarial training, etc. This would give the user a more comprehensive and clearer perspective.

---

### Official Review · Reviewer_G63v · 2024-11-08

**Soundness:** 4
**Presentation:** 4
**Contribution:** 3
**Rating:** 8
**Confidence:** 3

**Summary:**

This paper proposes a new benchmark to test LLM capabilities: whether they can generate adversarial attacks to proposed adversarial defenses. The authors crawled arXiv and filtered the papers to find adversarial defense methods with easily reproducible code. Current state-of-the-art LLMs were tested on this new benchmark, and do not perform well.

**Strengths:**

The authors propose a novel benchmark to test the capabilities of LLMs. The proposed task is a real-world research/security setting, where the "correct" answer may not even be known, and yet there is a quantitative measurement that can be extracted to evaluate the abilities of the LLM. As a result, this provides a benchmark which may still be useful even if a model has surpassed human level performance in this domain. The authors are upfront about the many limitations of the benchmark.

**Weaknesses:**

- The benchmark is evaluated in a limited setting. Even though to be successful, the model must be proficient in several different domains, the scope of the task is fairly limited.
- As mentioned, benchmark contamination is a potential issue, especially when considering the use of this benchmark well into the future.
- The benchmark does not appear to provide a meaningful continuous measure for current LLMs ability to generate novel attacks. Instead, it seems limited to whether they can successfully implement a known attack, as all of their current limited success on the benchmark is due to benchmark contamination.

**Questions:**

- As the models are currently not producing novel adversarial attacks, what does this benchmark measure that is not covered by existing benchmarks? Is the utility of this benchmark primarily for much more capable LLMs?
- How often do you expect to update the benchmark? If it is every time a new attack to a defense is published, then would this make it difficult to compare models trained at different points in time? If not, then might the benchmark provide an illusion of progress, even if it is just due of benchmark contamination?

---

> ### Author Response · Authors · 2024-11-22
> **Response to Review**
>
> We thank the reviewer for their time and valuable feedback. We address your comments below:
>
> > Benchmark is limited
>
> As models become more powerful, benchmarks must become be highly specialized to measure advanced capabilities and provide granular signal in narrow domains. Our benchmark includes a wide range of adversarial defenses with very different implementations and requires very different abilities (research skills, coding, debugging, tool use, etc.).
>
> > Benchmark contamination
>
> As we mention above, dataset contamination is a problem for any benchmark. We have taken multiple steps to measure the degree to which this is the case, and believe that it is not a significant problem at present.
>
> > Does not provide a continuous measure.
>
> We do not think this is the case. As we motivate in our paper, one of the advantages of our benchmark is that intermediate solutions will be measured (e.g. the model succeeds for 20% of images) providing a smooth signal of progress.
>
> > What does this benchmark measure that is not covered by existing benchmarks?
>
> Our benchmark is designed to measure the intersection of two capabilities:
> (1) The ability of models to perform sophisticated computer security attacks
> (2) The ability of models to interact with "real world" code that was not designed initially for evaluation purposes.
>
> Current benchmarks do not measure either of these. Existing security benchmarks typically are CTF-like and measure the ability of models to play security games that model what real world attacks look like (but are not real world attacks). And most coding benchmarks today are very clean "implement-this-one-function" benchmarks that do not handle challenging code. SWE-bench, which a significant step in this second direction, still has well-specified changes that need to be applied to a codebase; whereas we require sophisticated reasoning of the entire codebase to first reproduce and then break the proposed defense.
>
> > How often do you expect to update the benchmark?
>
> We intend to release our dataset initially containing as many defenses as we can up to the publication date. Since submitting our paper we have introduced a further 5 defenses, including the "Ensemble Everything Everywhere" defense that has received attention for making a new SoTA claim. We will update the benchmark as it becomes appropriate to do so (e.g., the accuracy on the updated benchmark is not nearly-perfectly correlated with the accuracy on the prior version).

---

> > ### Comment · Reviewer_G63v · 2024-11-26
> >
> > Regarding the continuous measure comment, the differentiation provided for current models seems fairly binary (see Table 2). This does not rule out a more meaningful continuous measure for future models.
> >
> > Thanks for your response. I'll maintain my rating.

---

### Author Response · Authors · 2024-11-22
**Overall comments**

We thank all the reviewers for their feedback and appreciate that they perceive our work as a “timely” and “clever idea for an agentic benchmark”. We would like to make two broad clarifications.

> Designing the best agent for this task is outside the scope of this work.

Several reviewers have mentioned that there is room for improvement in our agent. We agree. Benchmark papers are not expected to introduce highly capable specialized solutions to solve them, see, e.g., the ImageNet, MMLU, MATH, or MMMU papers each of which only introduce the problem and not the answer. Instead, what we do falls squarely in line with what these prior papers do: establish what can be done using current methods out-of-the-box. In this way, our agent is meant to demonstrate that (1) this benchmark is tractable, but (2) it is not trivially solved.

> No benchmark is free from data contamination.

Some reviewers have raised concerns about data contamination. It is essentially impossible to design a benchmark without some degree of contamination. (And even if there were to be absolutely zero contamination, any popular benchmark eventually becomes contaminated.) At least for now, most of the defenses covered in the benchmark do not have publicly available exploits. But even for those that do, we have found that we have found that if we provide the corresponding paper in context, current models can not even make use of this to succeed more often.

We believe the reason contamination is less impactful here is because when, e.g., MMLU is contaminated the answer is trivial (just regurgitate "The answer is C"). But for our case, even if the "answer" is online, the agent still must apply this knowledge to write new code to break the implementation we have. Thus, the impact of contamination, even if it is present, is smaller here.

Handling dataset contamination in LLMs that train on essentially all Internet knowledge is an open problem for any dataset, and we beleive we have done as much as is possible.

---

### Meta-Review · Area_Chair_zJLf · 2024-12-17

**Metareview:**

This paper introduces a new benchmark whereby LLMs are evaluated on their ability to break adversarial defenses.  The reviewers are conflicted about this paper.  At a high level, I think the scope of this paper is narrow, and I think the fact that the benchmark entangles multiple factors like knowledge of existing codebases and ability to reason about what might break a particular defense is a bug and not a feature of a benchmark.  Due to the multiple negative reviews (and accounting for the fact that reviewers did interact during the rebuttal period), I recommend rejection for this paper.  Nonetheless, I encourage the authors to keep improving their work.

**Additional Comments On Reviewer Discussion:**

Reviewers did interact with the authors during the rebuttal period, but they largely maintained their accept/reject opinions.  One reviewer supported the paper, but I did not find their points compelling.

---

### Decision · Program_Chairs · 2025-01-22

Reject